# The Effect of A Whey-Protein and Galacto-Oligosaccharides Based Product on Parameters of Sleep Quality, Stress, and Gut Microbiota in Apparently Healthy Adults with Moderate Sleep Disturbances: A Randomized Controlled Cross-Over Study

**DOI:** 10.3390/nu13072204

**Published:** 2021-06-27

**Authors:** Anne Schaafsma, Leonard Mallee, Maartje van den Belt, Esther Floris, Guus Kortman, Jouke Veldman, Daan van den Ende, Alwine Kardinaal

**Affiliations:** 1FrieslandCampina, 3818 LE Amersfoort, The Netherlands; leonard.mallee@frieslandcampina.com (L.M.); jouke.veldman@frieslandcampina.com (J.V.); 2NIZO, Nutrition & Health, 6710 BA Ede, The Netherlands; maartjevandenbelt@gmail.com (M.v.d.B.); estherfloris@hotmail.com (E.F.); guus.kortman@nizo.com (G.K.); alwine.kardinaal@nizo.com (A.K.); 3Philips Research, 5656 AE Eindhoven, The Netherlands; daan.van.den.ende@philips.com

**Keywords:** sleep, stress, microbiota, galacto-oligosaccharides, GOS, whey protein, tryptophan, skimmed milk powder

## Abstract

People experiencing sleep problems may benefit from nutrients supporting serotonin metabolism and stress reduction. We studied the effect of a dairy-based product (DP) containing protein, galacto-oligosaccharides, vitamins and minerals, on sleep quality, stress, and gut-microbiota. In a cross-over RCT (three weeks intervention; three weeks washout), adults (*n* = 70; 30–50 y) with sleep disturbances (Pittsburgh Sleep Quality Index (PSQI) ≥ 9) consumed products 1 h before bed-time. Sleep quality (PSQI) was measured weekly, stress at base- and end-line (Depression Anxiety Stress Scale and saliva cortisol). Fecal samples were collected in the 1st intervention period only. Compared to placebo (skimmed milk), PSQI was only lower at day 14 in the 2nd intervention period in intention-to-treat (ITT) (*p* = 0.017; *n* = 69) and per-protocol (PP) (*p* = 0.038; *n* = 64) analyses. Post-hoc analysis (modified-PP: n=47, with baseline PSQI ≥ 9, and endline day 14), however, showed a decrease in PSQI (−1.60 ± 2.53; *p* = 0.034). Early morning saliva cortisol decreased versus placebo (*p* = 0.045). Relative abundance of *Bifidobacterium* increased (*p* = 0.02). Redundancy analysis showed an inverse relationship between baseline microbiota composition and baseline PSQI (*p* = 0.046). Thus, although DP did not improve sleep quality in ITT and PP populations, it did in the modPP. DP reduced salivary cortisol and stimulated *Bifidobacterium*, which possibly is important for sleep improvement.

## 1. Introduction

A good night of sleep is beneficial for overall health and well-being [1,2,3]. However, insufficient sleep is prevalent across various age groups, often unrecognized, under-reported, and causing high economic costs. It increases the risk of several metabolic derailments (e.g., diabetes mellitus, overweight) and changes behavior (e.g., less conscious, depression) [1]. Lifestyle is an important factor in sleep quality. Late evening use of smart phones, electronic devices [1], exposure to blue light [4], caffeine containing drinks [5], and possibly stress in particular [6] are important factors of sleep disturbances.

Sunlight is a powerful timing cue of the human circadian rhythm. Although this biological clock regulates timing of sleep, it is not responsible for sleep itself [7,8]. The neuropeptide orexin is recognized as an important on/off switch of sleep. Stimulation of orexin synthesis induces waking up. This synthesis is stimulated by, among others, low levels of blood glucose, insulin, serotonin, and gamma-aminobutyric acid (GABA), as well as by increased stress and acidification of cellular environment [9]. The involvement of these stimulating factors suggests that nutrition can play an important role in sleep quality.

Cow’s milk derived whey protein is known to be a good source of tryptophan (Trp). This amino acid is the indirect precursor of serotonin (relaxing neurotransmitter) which is used for melatonin (sleep supporting hormone) production [10]. The availability of Trp for serotonin production depends on its ratio with other large neutral amino acids (competitive uptake in the brain) [10]. Furthermore, availability is decreased in case of stress and inflammation when more Trp is pulled into the kynurenine pathway [11]. Considering this link between stress and sleep, the ability of a tryptic hydrolysate of bovine αs1-casein (150–300 mg/day; Lactium^®^) to reduce stress levels in adults is of interest [12,13,14]. The same accounts for the indicated effect of galacto-oligosaccharides (GOS; Biotis ^TM^ GOS) on anxiety [15]. Both the tryptic hydrolysate and GOS possibly affect the GABA pathway in the brain via intestinal GABA receptors: ‘directly’ by the tryptic hydrolysate and indirectly by GOS via stimulation of GABA-producing gut bacteria [16,17,18,19,20]. Communication between intestinal microbiota and the brain on stress behavior of the host might take place via the vagus nerve, gut hormone signaling, the immune system, Trp metabolism, and microbial metabolites (such as short chain fatty acids, GABA, and allopregnanolone precursor hyodeoxycholic acid) [11,21,22].

To support the serotonin synthesis and to preserve serotonin, vitamin D is important [23]. Other elements that are associated with sleep and/or depression are magnesium and zinc [24]. Magnesium is an agonist of GABA [25,26], and appears to play a role in controlling pH of the extra-cellular fluid via magnesium-dependent transmembrane pumps of bone cells that replace hydrogen ions in the extra-cellular fluid by potassium ions [27]. Zinc, among others, can modulate the activity of several neurotransmitter receptors [28], including adenosine [29], dopamine [30], and serotonin [31].

In other words, a combination of aforementioned components in adequate amounts should theoretically support sleep quality, preferable in combination with lifestyle improvement, and appears to be a safe approach.

In the present explorative cross-over study, a whey protein-GOS-based supplement enriched with tryptophan, a tryptic casein hydrolysate, magnesium, zinc, niacin, vitamin B_6_, and vitamin D_3_ was consumed every evening for three weeks by apparently healthy Dutch adults with mild-moderate sleep disturbances. Skimmed milk was used as a control product. The primary objective was to get insight into the effect of the study product on sleep quality as measured by the Pittsburg Sleep Quality Index (PSQI) questionnaire. Secondary outcomes were the effects on stress/anxiety/depression (Depression Anxiety Stress Scale-42 (DASS-42)), early morning saliva cortisol levels, and sleep pattern as measured by a sleep tracker (SmartSleep, Philips, Eindhoven, The Netherlands). Tertiary outcome was the effect of the study product on gut microbiota composition.

## 2. Materials and Methods

### 2.1. Study Design

The study was designed as an explorative double-blind randomized placebo-controlled cross-over study in 70 apparently healthy Dutch adults. Candidates were recruited via advertisements on social media and in regional newspapers, and posters mounted in public buildings (via Link2Trials, https://www.link2trials.com/clinical-trials (accessed on 25 June 2021)). Potential candidates received study information and were invited for a study information meeting at the study center NIZO (Ede, The Netherlands). Following signing of the informed consent (IC) form, subjects were screened for eligibility using an online health and lifestyle questionnaire, addressing the inclusion and exclusion criteria, and the PSQI questionnaire.

Final inclusion of subjects, by the principal investigator, were randomized (blinded, by the data manager, all at the same time) using a Research Manager (Research Manager, Deventer, The Netherlands) generated randomization list, stratified for gender and age (<39 years or ≥40 years) to start with either the dairy-based product (DP) or placebo, each for 3 weeks and a washout period of 3 weeks (intervention periods: 2–23 December 2019 and 13 January till 3 February 2020). All subjects were instructed on study procedures, such as questionnaires and collection of biological materials before the start of the study. At the baseline visit, body weight and height were measured, and instructions were given on study product intake and on other baseline assessments. During each intervention period, a daily questionnaire on bedtime and wake-up time, lifestyle habits, sleep characteristics, and fitness and mood at waking up had to be filled in online. The PSQI and product tolerability questionnaire were completed online on a weekly basis (day 0, 7, 14, and 21). Five saliva samples were collected by the subject during each intervention period at the start and after 21 days of product consumption, starting at waking up and after that every 15 min. Samples were kept in the home-freezer. Sleep tracker (SmartSleep, Philips, Eindhoven, The Netherlands) measurements took place for five consecutive nights, before product consumption, of each intervention period and during the last 5 nights of each period. Fecal samples were collected by the subject during the 1st intervention period only, at days 0 and 21, and stored in the home-freezer. The washout period was used to supply new products for the next intervention period whereas remaining study products, emptied packages, and the frozen saliva and fecal samples of the 1st intervention period were collected and returned to NIZO. At the end of the study, participants visited the study center again for the measurement of body weight, the return of the SmartSleep, and to bring the frozen saliva samples of the 2nd intervention period as well as remaining study products and emptied packages. All visits took place at NIZO (Ede, The Netherlands).

The study was conducted in accordance with the Declaration of Helsinki, and approved by the Dutch medical research ethics committee METC-WU, dossier number NL70673.081.19, September 2019, and registered in the Netherlands Trial Registry (identifier: NL7919).

### 2.2. Subjects

Seventy apparently healthy men and women, with sleep problems met all of the following criteria at inclusion: age 30–50 y, BMI 19.5–25 kg/m^2^, PSQI ≥ 9. Potential subjects were excluded in case of use of medicines or supplements which affected sleep, diagnosed sleep diseases, being allergic or intolerant to ingredients of the products, being involved in shift working, having a serious risk for jetlag (intercontinental flight and from a time-zone with >3 h difference ≤1 week before an intervention period), being pregnant or breastfeeding, being treated by a psychologist for sleep or burnout, diseases of the respiratory tract that cause serious sleep issues (as assessed by the study physician), use of soft and hard drugs during the study period, a history of medical or surgical events that may significantly affect the intestine and/or digestion, a mental status that is incompatible with the proper conduct of the study, alcohol consumption for men: >28 units/week and >4/day or for women: >21 units/week and >3/day, reported weight loss or weight gain of >3 kg in the month prior to pre-study screening, or intention to lose weight during the study period, reported slimming or medically prescribed diet, being employed by FrieslandCampina Research, NIZO, or Philips Research (or their partners and their first and second degree relatives), and (self-reported) peri- or postmenopausal women.

### 2.3. Products

Composition of both study products, packed in identical white sachets, is shown in Table 1. To mask taste differences, products were vanilla flavored and sweetened with sucralose. Products had to be dissolved in 150 ml of lukewarm water, and were consumed once daily, about 1 h before going to bed. Emptied sachets had to be collected and returned for a check on compliance. In the daily questionnaire, subjects had to report whether the product was indeed consumed, and whether this was done at the correct time. Participants received per intervention period a box containing sufficient sachets (with 1 spare sachet) for the intervention period, numbered 1–1 to 1–21 for the 1st intervention period, and 2–1 to 2–21 for the 2nd intervention period. Blinding took place by using three 3-character codes for each of the study products. Only the product developer had access to decoding information. Next to that, only an independent researcher at NIZO (not involved in the study performance) had access to group allocation information.

### 2.4. Methods

Lifestyle habits and sleep characteristics were assessed by a daily online questionnaire reporting bedtime and wake-up time, lifestyle habits (servings of caffeine and alcohol containing drinks, TV/computer/mobile phone use and/or book reading 30 min before going to bed), and visual analogue scales (using 5 emoticons) for fitness and mood at waking up. Average values for every 7 days were evaluated.

Sleep quality was registered by an online PSQI questionnaire at days 0, 7, 14, and 21 of each intervention period. This validated questionnaire (https://eprovide.mapi-trust.org/instruments/pittsburgh-sleep-quality-index (accessed on 25 June 2021)) contains 19 self-rated and 5 bed partner/roommate (if available) rated questions. Per time point, a total PSQI score was calculated based on 7 component scores each of which has a range of 0–3 points [32].

The DASS-42 questionnaire was completed online at days 0 and 21 of each intervention period, containing 42 questions with component scores for depression, anxiety, and stress [33].

Cortisol levels were determined in 5 consecutive early morning saliva samples per participant starting at waking-up and after that every 15 min, at days 0 and 21 of each intervention period, using LC-MS/MS (6470 Angilent, Brightlabs, Venlo, The Netherlands) Samples were collected in salivettes (Sarstedt, Nümbrecht, Germany) and stored in the freezer until transport to the study center. Upon arrival, the salivettes were frozen at −20 °C until analysis.

Sleep patterns (sleep efficiency, sleep duration, sleep onset, rapid eye movements (REM), non-rapid eye movements (NREM), and wake up after sleep onset (WASO)) were measured during the night using a head-band sleep tracker (SmartSleep, Philips, Eindhoven, The Netherlands) for 5 consecutive days (only the first time of SmartSleep use included an additional day for habituation) before product intake and during the last 5 days (days 17–21) of each intervention period. Data were averaged per 5 days, or less when measurements per night were not complete. Incomplete readings were defined as: total sleep duration + WASO + time to sleep onset <270 min. Read out of all SmartSleep study parameters were done blinded by Philips Research at the end of the study. The SmartSleep is a consumer biofeedback EU class 1 radio product and conforms to the applicable general standard EN 60335-1:2012/A13:2017.

Body weight and height (only at the start) were measured and recorded during the visits to the study center. Weighing took place without shoes and heavy over-clothes using a calibrated personal scale (Kern MPE250K100PM, Kern & Sohn GmbH, Balingen, Germany). Measurement of height was performed with a calibrated mobile stadiometer (Seca 213, Seca, Hamburg, Germany). Based on weight and height, the BMI (kg/m^2^) of each participant was calculated. 

Microbiota was studied in spot fecal samples (collected at the start and end of intervention period 1). Subjects had to collect approximately 5–10 g of stool in a 50 mL sample container with a screw cap. Collected samples were kept at −20 °C for a maximum of 4 weeks until transport to the study center by courier. DNA was isolated from thawed and homogenized fecal samples as previously described (disruption of microbial cells, and extraction and purification of nucleic acids with a magnetic beads procedure) [34]. Barcoded amplicons from the V3–V4 region of 16S rRNA genes were generated using a 2-step PCR and according to previously described methods [34]. On 14 samples (13 d0 samples and 1 d21 sample) and control samples, we applied a low-biomass PCR protocol to be able to generate 16S rRNA gene amplicons using the same primer set. The PCR amplification mixture contained: 4 μL 10× diluted fecal DNA, and 16 μL master mix [0.2 μL Phusion DNA Polymerase (2 U/μL; ThermoFisher Scientific, Waltham, MA, USA), 4 μL Phusion HF buffer (5×), 0.4 μL dNTP mix (10 mM each)], 0.1 μL (100 μM) of forward primer, 0.1 μL (100 μM) of reverse primer, 0.6µL DMSO, and 10.6 μL sterile water (total volume 20 μL). PCR conditions were: 95 °C for 2 min, followed by 30 cycles of 95 °C for 20 s, 55 °C for 20 s and 72 °C for 1 min. This was followed by a final extension for 10 min at 72 °C. The approximately 500 bp PCR amplicons were purified using the MSB Spin PCRapace kit (Invitek Molecular, Berlin, Germany).

For the library PCR step with sample-specific barcoded primers, purified PCR products were shipped to BaseClear BV (Leiden, The Netherlands). PCR products were checked on a Bioanalyzer (Agilent) and quantified. This was followed by multiplexing, clustering, and sequencing on an Illumina MiSeq with the paired-end (2x) 300 bp protocol and indexing. The sequencing run was analyzed with the Illumina CASAVA pipeline (v1.8.3) with de-multiplexing based on sample-specific barcodes. Sequence reads of too low quality (only passing filter reads were selected) and reads containing adaptor sequences or PhiX control were discarded from the raw sequencing data. On the remaining reads, a quality assessment was performed using FastQC version 0.10.0. (http://www.bioinformatics.babraham.ac.uk/projects/fastqc/ accessed on 25 June 2021). 

16S rRNA gene sequences were analyzed using a workflow based on Qiime 1.8 [35]. On average, 45,666 (range 22,417–56,586) sequences were obtained per sample to define their taxonomic profiles. We performed operational taxonomic unit (OTU) clustering (open reference), taxonomic assignment, and reference alignment with the pick_open_reference_otus.py workflow script of Qiime, using Uclust as clustering method (97% identity) and GreenGenes v13.8 as reference database for taxonomic assignment. Reference-based chimera removal was done with Uchime [36]. The RDP classifier version 2.2 was performed for taxonomic classification [37]. Quality control was performed on all samples and low-biomass PCR protocol was accounted for in subsequent analysis where necessary.

### 2.5. Statistics

Sample size. Based on the study of Saint-Hilaire (2009), the decrease (i.e., improvement of sleep) in PSQI score following a 28-day treatment with 150 mg tryptic casein hydrolysate was −3.5 with an estimated SD of 3.98 (based on an unpublished consumer study: supplemental white paper), whereas the placebo group showed a decrease in PSQI score of −2.5. [12]. Based on the cross-over design of the present study and a stricter selection of participants (PSQI at inclusion of ≥9), a smaller SD (3 instead of 3.98) was expected and was used in the calculation. The treatment difference between the study groups was estimated to be at least 1.5 units PSQI, which is higher than in the Saint-Hilaire study, but realistic since the present DP is more than only tryptic casein hydrolysate. Using an SD of 3 and estimated difference in PSQI outcome of 1.5, a significance level (adjusted for sidedness) of 0.025, and a power of 0.8, a total of 65 participants should be evaluated. To correct for some drop-outs, we included 70 persons—sample size tool for cross-over studies: http://hedwig.mgh.harvard.edu/sample_size/js/js_crossover_quant.html (accessed on 25 June 2021).

Data analysis sets. The intention-to-treat (ITT) data analysis set includes all subjects randomized to study products and with at least one measurement in place. The per protocol (PP) data analysis set includes all subjects of the ITT data analysis set, except those with major protocol violations (compliance to product consumption <80%, time of product consumption, time of measurements, use of sleep medication). However, it appeared posthoc that, at a baseline of the 1st intervention period, 17 subjects had a PSQI score of <9, although the screening outcome was ≥9. For unknown reasons, the PSQI value of these subjects improved in between screening and baseline. For that reason, a modified PP (modPP) data analysis set was created consisting of PP data minus these 17 subjects with improved PSQI-score. Primary analyses also indicated that the maximum effect of both products was present at day 14 (see results). For that reason, day 14 was added as end-line in the post-hoc data evaluation.

Data evaluation. For the primary study parameter, changes within and differences between study products were evaluated per intervention period as well as when intervention periods were taken together. Missing data were handled as missing, i.e., no imputation was done. For the primary outcome, mixed model analysis was performed with intervention period, time point, and interaction of study product by time point as fixed effects and subject as random effect for the change from baseline. In addition, General Linear Model Repeated Measurements (GLMR) with Bonferroni post-hoc test has been used to study within group changes when the parameters had more than two measurements in time. Carryover effect and treatment effect have (also) been studied using a statistical method specifically for cross-over design. In case a carry-over effect is indicated, taking intervention periods together may not provide a trustful outcome. In fact, in that case, only the results of the first intervention period should be used [38]. For the evaluation of baseline and endline values (either day 7, 14 or 21), or time-point differences between groups, a parametric (T-test) or non-parametric test (Mann–Whitney U-test) was used. Normality was studied with the Shapiro–Wilk test. For an ‘effect’ (any decrease in PSQI) or ‘no effect’ (no change or increase in PSQI) analysis, the McNemar test was used as described by J.W.R. Twisk [39].

For the microbiota data, statistical tests were performed as implemented in SciPy (https://www.scipy.org/ accessed on 25 June 2021), downstream of the Qiime-based workflow. Between-group differences in alpha diversity (PD_whole tree) were tested with the Mann–Whitney U test, as implemented in Graphpad Prism 5.01 (San Diego, CA, USA). Between group differences of single taxa were assessed using the Mann–Whitney U test with FDR correction for multiple testing. Comparisons of the targets of our primary interest (*Lactobacillaceae, Bifidobacteriaceae, Enterobacteriaceae*) were not corrected for multiple testing.

To compare microbiota compositions, multivariate redundancy analyses (RDAs) were performed as assessed by 16S rRNA gene sequencing in Canoco version 5.12 using default settings of the analysis type “Constrained” [40]. Relative abundance values of OTUs or genera were used as response data, and metadata as explanatory variable. For visualization purposes, families or genera, rather than OTUs, were plotted as supplementary variables. Variation explained by the explanatory variables corresponds to the classical coefficient of determination (R2) and was adjusted for degrees of freedom (for explanatory variables) and the number of cases. Canoco determines RDA significance by permutating (Monte Carlo) the sample status. Per time point and sample set, confounding factors were first identified by RDA. Statistically significant confounders were included as covariates in subsequent analyses. Hence, partial RDA was employed to correct for covariance where relevant, covariates were first fitted by regression and then partialled out (removed) from the ordination.

Outcomes were evaluated per treatment and between treatment-groups (ITT).

Statistical significance was tested 2-sided with a statistical significance level of 0.05. In addition, *p*-values >0.05 and <0.10 are considered to indicate a trend towards an effect. Analyses were performed, except for microbiota data, using SAS software version 9.4 or higher (SAS Institute, Cary, NC, USA), and/or IBM SPSS Statistics, Version 24 (IBM Corp., Armonk, NY, USA) 

## 3. Results

### 3.1. Subjects

Subject flow and data analysis sets are shown in Figure 1, and characteristics of participants are reported in Table 2. Following advertisements, 888 persons indicated their interest of whom 461 were invited for the information meeting at the study center. Upfront, 98 withdrew, whereas 213 subjects did not show up.

### 3.2. Sleep Quality

#### 3.2.1. PSQI-Score

PSQI values in the ITT and PP populations were normally distributed at days 0, 7 and 21, but not at day 14 (*p* < 0.05, Shapiro-Wilk). In the modPP group, data at all days were normally distributed. No carry-over effect was indicated.

Changes in absolute PSQI scores (and sub-scores) were not different between placebo and DP in both the ITT and PP population (taking both periods together) at any time point (Table 3). Only in period 2, and at day 14, DP had a lower PSQI score than placebo (ITT *p* = 0.017, PP *p* = 0.038) (Appendix A). In the modPP population, at day 14, the absolute PSQI score was lower, and the change in PSQI-score was bigger [−1.60 ± 2.53 (14.7%) versus −0.30 ± 3.28 (2.8%)], for DP as compared to placebo (*p* = 0.034) (Figure 2). In line with this, a treatment effect was found (*p* = 0.027) by using the method of Wellek et al. [38]. Delta PSQI did not differ between DP and placebo at days 7 and 21. With regard to the PSQI sub-scores (not presented), only sleep onset latency showed a trend of being shorter with the DP as compared to the placebo (modPP, *p* = 0.08, effect difference: 15%, 95% CI: −1.5–30.3%).

Within treatment groups, and for ITT and PP populations, no changes were found in the course of time in either delta PSQI or absolute PSQI score during the intervention. Only in the ITT population the PSQI-score did tend (*p* = 0.072) to be lower for DP, mainly caused by the change in PSQI-score between baseline and day 14 (*p* = 0.068) (Table 3). With regard to the modPP, only DP showed an improvement in PSQI score in the course of time (Table 3). Comparing each time point with baseline (information not shown), DP showed a significant decrease in PSQI at every time point: 0–7 days (*p* = 0.037), 0-14 days (*p* < 0.001) and 0–21 days (*p* = 0.006). No effects were found for the placebo at any time point (*p* > 0.362).

#### 3.2.2. SmartSleep

In total, 23 persons dropped out because of incomplete readings of the SmartSleep (*n* = 10; loss of sensor connection, depleted battery or because the headband was removed during the night) or not having enough datapoints (*n* = 13): 46 persons were evaluated. All data from the SmartSleep were normally distributed (Appendix A). No carryover effect was indicated. This method showed a treatment effect (*p* = 0.049) for REM sleep: being 10 min longer (about 9% of total REM sleep) with the DP due to a decline in REM in case of the placebo. No treatment effects were found for any other SmartSleep parameter.

### 3.3. Daily Lifestyle Questionnaire

Daily online reporting of bedtime and wake-up time, lifestyle habits (servings of caffeine and alcohol containing drinks, TV/computer/mobile phone use and/or book reading 30 min before going to bed), and visual analogue scales (using 5 emoticons) for fitness and mood at wake-up time did not change during the study, and were not different between the treatment groups. Baseline values are reported in Table 2.

### 3.4. Depression–Anxiety–Stress Outcomes

#### 3.4.1. DASS-42 Questionnaire

DASS-42 scores in the ITT and PP populations were non-parametric (*p* = 0.006), whereas they were normally distributed in the modPP group. 

No differences were found for total DASS-42 score, or any of the sub-scores, between DP and placebo in ITT, PP, or modPP populations (Table 4). However, a carryover effect was indicated for the ITT population (trend for the PP population, not for modPP), for at least the anxiety sub-score (*p* = 0.022). Median (min-max) DASS-42 score in the ITT population at baseline was 14 (0–46), indicating a mild (score: 10–13) to moderate (score: 14–20) affected population [41,42].

Within treatments, placebo showed improvements in stress sub-score (Table 4) in ITT, PP and modPP, and in total DASS 42 score in the ITT population. The DP showed a trend towards improvements in total score in PP and modPP (*p* = 0.055 and *p* = 0.065, respectively) and depression sub-score in modPP (*p* = 0.067). Of interest, trends for improvement by DP were in particular related to effects in the 1st intervention period, whereas for placebo effects this was the case for the 2nd intervention period (Appendix A).

#### 3.4.2. Salivary Cortisol

Cortisol data (ITT, PP, modPP) were non-parametric distributed at all time points and/or for each treatment. Cortisol levels were lower with DP in the first saliva taken at day 21, as compared to placebo (ITT *p* = 0.045, modPP *p* = 0.033, and PP *p* = 0.059) (Table 5, all cortisol outcomes in Appendix A). Speed of increase in cortisol levels, during the first 30 min after waking-up, tended to be lower (*p* = 0.057, ITT population) with the DP, however when correcting for outliers (based on boxplot analysis), the p-value increased to 0.129. A trend towards a higher cortisol level at awaking in the PP population after taking the placebo product for 21 days was unexpected (*p* = 0.098).

### 3.5. Gut Microbiota

Fecal samples were only collected in the first intervention period at baseline (IP *n* = 35, Placebo *n* = 34) and after three weeks of intervention. In redundancy analysis (RDA), gender, age, and BMI did not affect the results at either time point. Gut microbiota profiles in the ITT, as assessed by RDA, and alpha diversity were not significantly different between groups at day 21. Nevertheless, within the DP group, there was a significant difference between d0 and d21 (variation explained 3.0%; *p* = 0.004, Appendix A), in which d21 was associated with e.g., a higher relative abundance of *Bifidobacterium*. This was not the case in the placebo group (variation explained; 0% *p* = 0.6). The increase in *Bifidobacterium* (taxon of primary interest) at d21 was different from the unchanged abundance in the placebo group (*p* = 0.02) (Figure 3). No effects were seen in the relative abundance of *Lactobacillus* and *Enterobacteriaceae.* A low relative abundance of *Bifidobacterium* at baseline was a predictor for sleep improvement, irrespective of treatment. This was significant for the whole ITT group (R = 0.33, *p* = 0.0065), and borderline statistically significant for the separate products (DP: R = 0.33, *p* = 0.06; Placebo: R = 0.35, *p* = 0.04).

### 3.6. Product Tolerability and Adverse Events

DP and placebo did not show differences in product tolerability: flatulence, nauseous or bloated feelings. 

In total, 19 and 21 AEs were reported for DP and placebo, respectively. Two SAEs (allergy to horses, kidney stone) were not product related. Possible product related AEs were cramps and diarrhea (placebo: 1), diarrhea and vomiting (placebo: 1), firm feces (placebo: 1), nausea (DP: 1), heartburn (DP: 1), and hunger feelings (DP: 1). Five additional AEs (itching, irritation, sore ears, headache) were related to the use of the SmartSleep headband (DP: 4; placebo: 1). Not-product related adverse events included headache, migraine, common cold, influenza, stomach-ache, inflammation of upper respiratory tract, back-shoulder pain, and food poisoning.

## 4. Discussion

In the present study, the primary goal (an improvement in PSQI-score by DP as well as a lower PSQI-score as compared to placebo) was not reached in ITT and PP populations. Only in the 2nd intervention period the DP did result in a greater drop in PSQI score at day 14. With regard to other objectives, DP resulted in lower early morning salivary cortisol levels (as compared to placebo: −23% in modPP, −16% in PP, and −15% in ITT), and in higher relative abundance of fecal *Bifidobacteriaceae*. No differences were found in the DASS-42 total- and sub-scores. The SmartSleep headband (in a sub-population of ITT) showed a 9% decrease in REM sleep with placebo but no change in case of DP. 

Looking at baseline PSQI data post hoc, it appeared that 17 persons did not fulfill the screening criterion PSQI-score ≥9 anymore. Next to that, ITT and PP analysis indicated that effects were greater at day 14 of intervention instead of day 21. Based on this, a modified PP population (modPP, *n* = 47) was put in place. In this population, the absolute PSQI score was lower and the delta-PSQI was bigger with DP as compared to placebo at day 14. Furthermore, as compared to baseline, the decrease in PSQI score by DP was significant at every time point (7, 14, and 21 days), with no effects in case of placebo. In addition, in the course of time, the absolute PSQI score improved, which was in particularly due to an improvement from day 0–14. Of the PSQI sub-scores, only ‘sleep onset’ tended to be shorter from the placebo group (*p* < 0.09). These modPP results indicate that the PSQI score at baseline makes a big difference in outcome.

Although the effect size (−1.6 ± 2.5) of the DP on PSQI-score in the modPP population is small, it is in line with two other studies (−1.3 and −1.8; with supplements containing at least the tryptic casein hydrolysate) using the PSQI score as outcome [17,43]. Of interest, these two studies did not find a difference with placebo, in contrast to the present study. Furthermore, the 14.7% improvement in PSQI score, as shown with DP, is within the effect size range (13–19%) as reported for acupressure [44], and in line with 14.2% improvement in healthy adults supplemented with 200 mg/d L-theanine for four weeks [45]. In an uncontrolled pre-trial consumer study (Supplemental white paper), however, the PSQI effect size of the DP was bigger (−3.35: 11.20 ± 0.56 to 7.85 ± 0.55; *p* < 0.001; *n* = 54), and in line with the study of Saint Hilaire et al.: −3.5 [12]. Furthermore, it is hypothesized that the mild to moderate stress level in the present study might have played a role in the lower effect size, since the DP addresses stress in particular.

Stress, in combination with anxiety and depression, was measured using the DASS-42 questionnaire. Median (min-max) DASS-42 score in the ITT population at baseline was 14 (0–46), which indicates a mild to moderate affected population [41,42]. This is supported by normal cortisol levels in early morning saliva samples. Unexpectedly, the placebo group showed an improvement in stress sub-score (in ITT, PP and modPP) due to significant changes in intervention period 2 only. This finding could relate to an indicated carryover effect [38]. No statistical effect of either product in intervention period 1, though a trend (*p* = 0.090) for a lower DASS-42 score in case of DP, suggests that a short-term effect is not expected in the tested target group. In addition, ‘no effect’ by DP in intervention period 2 indicates that the placebo in intervention period 1 did not have a ‘delayed’ effect on DASS-42 outcomes. Thus, an effect of placebo in intervention period 2 is unlikely but may reflect a carryover effect. A possible effect of DP on stress is supported by lower cortisol levels in early morning saliva samples, as compared to placebo. Based on these outcomes and indications, a wash-out- as well as an intervention-period of 3 weeks might not be long enough to show an effect of the DP on at least the DASS-42 total score.

A possible effect on stress/anxiety/depression might be explained via at least three routes: tryptophan-serotonin, casein tryptic hydrolysate-GABA, and intestinal *Bifidobacteriaceae* [10,11,13,46,47,48,49,50,51,52]. Whereas the first two routes should result in short term responses, the microbiota route needs more time. The right bacteria, such as *Bifidobacteriaceae*, have to grow before active metabolites (e.g., GABA) are produced or intestinal production of serotonin and melatonin is stimulated in a sufficient way [10,11,20,53]. In an animal study, administration of a prebiotic diet (including GOS) attenuated the formation of metabolites that could act as neuroactive steroids potentiating GABAergic inhibition [22]. In addition to these animal studies and in line with the present study, daily administration of 5.5 g of GOS (Bimuno-GOS) during a three-week period reduced the salivary cortisol awakening response in healthy people [54], and daily administration of GOS (3.5 or 7 g) for four weeks stimulated growth of intestinal *Bifidobacteriaceae.* However, only the 7 g dosage was effective in reducing anxiety symptoms in a high anxiety target population of young adults [15,55]. Based on the above, and the DASS-42 and cortisol outcomes in the present study, it is hypothesized that the stimulation of *Bifidobacteriaceae* contributes to a possible beneficial effect of DP on stress/anxiety/depression parameters. This would be in line with our suggestion that *Bifidobacterium* at baseline appear to be inversely related with the possibility to improve sleep.

The use of the SmartSleep headband intended to provide objective information on sleep architecture and insight into time-to-fall asleep. However, due to data readout issues and incomplete readings (96 data points had to be removed; 8.3% of total data points), the ITT population shrank from 69 to 46 participants. Furthermore, since we did not introduce time-to-bed restrictions, as done in other studies, this parameter showed quite a broad range: from 9:00 p.m. to 4:00 a.m., and an irregular bedtime disturbs sleep architecture [56]. In the present study, only REM sleep proved to be different between DP and placebo. It was not affected by DP but decreased in case of placebo, which resulted in a modestly longer (about 9%, expressed in min) REM sleep with DP. Stabilization of REM sleep might be an effect of a stimulated Trp-melatonin pathway. Kunz et al. showed a positive effect of oral melatonin (3 mg; between 10:00 p.m.–11:00 p.m. for four weeks) on REM sleep in humans with disturbed REM sleep, as compared to placebo [57]. No effect of DP on the amount of REM sleep in the present study can possibly be explained by the fact that subjects had no abnormalities in their REM sleep. Furthermore, studies show differences in methodology (polysomnography versus SmartSleep ‘s single channel EEG). Of interest, a prebiotic (including GOS) stimulation of gut microbiota (resulting in less keton steroid), was also associated with improved REM sleep [22].

Limitations of this study are in particular the necessity to create a modified PP population to be able to show the effect of DP and its difference with placebo. The strongly reduced number of participants in the modPP was below the estimated sample size. In a future study, inclusion/exclusion criteria should not only be of importance/checked at recruitment but also at baseline. The second limitation is about shifting the end-line for the modPP population from 21 days to 14 days. In this way, a direct comparison with saliva, fecal, DASS-42, and SmartSleep outcomes was not possible since these parameters were only measured at baseline and after 21 days of intervention. We can also not exclude that Christmas holidays during the washout period had an effect on the 2nd intervention period. Finally, for at least the DASS-42 outcomes, a carryover effect cannot be excluded. For these kinds of parameters, the washout and intervention periods should likely be considerably longer (probably 6–8 weeks). In other words, a parallel study design might be preferred. Strengths of the study were the quality of performance, and several complementary outcomes (sleep quality, cortisol, and gut microbiota) that, in combination with the pre-clinical trial, all point to an effect of the DP.

## 5. Conclusions

The effect of DP on sleep quality (PSQI score) was not shown in the primary ITT and PP analyses. However, some participants (*n* = 17) showed spontaneous improvements in sleep quality from screening to first consumption of product. Taking out these persons (modPP), the present study did show a beneficial effect of DP over placebo on sleep quality after 14 days of intervention. Despite the low stress levels of most of the participants, DP also decreased early morning saliva cortisol as compared to placebo. Bifidobacteria increased only in the case of DP (ITT population). Together, the present study provides a good indication that the test product can relieve moderate sleep disturbances in adults with a maximum effect after day 14, but this needs confirmation.

## Figures and Tables

**Figure 1 nutrients-13-02204-f001:**
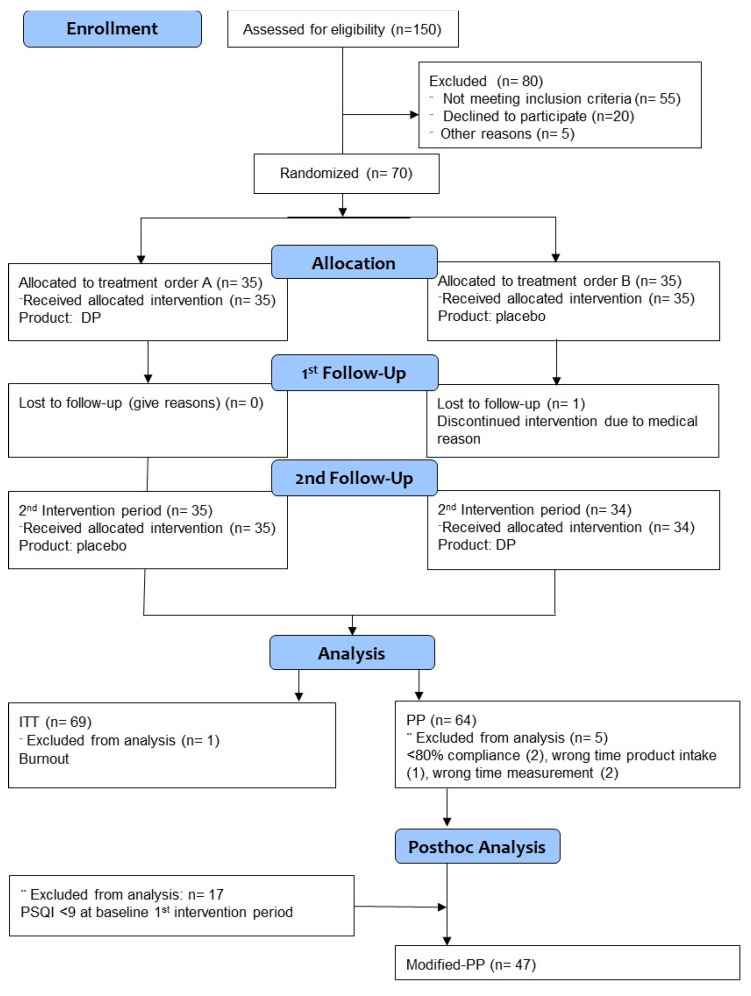
The recruitment–allocation–analysis flow diagram. DP, based on CONSORT 2010 but adapted for cross-over design: dairy-based test product; PSQI: Pittsburgh Sleep Quality Index.

**Figure 2 nutrients-13-02204-f002:**
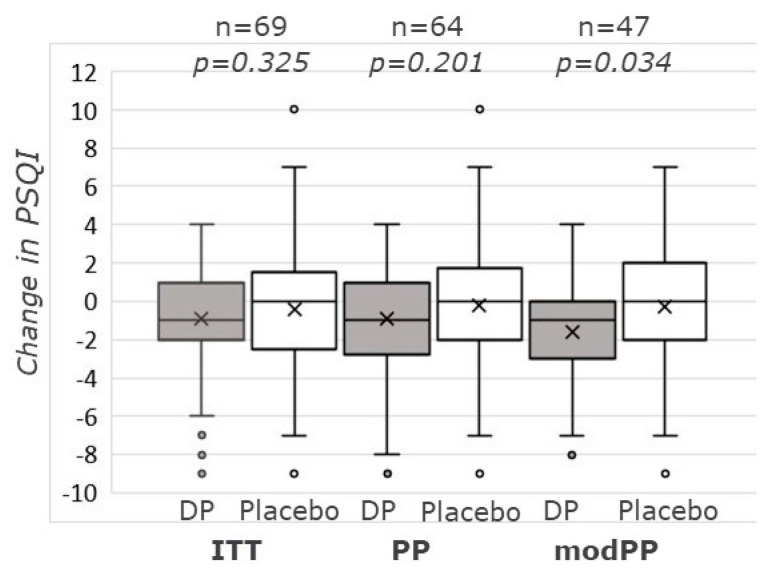
Changes in PSQI from baseline to day 14, for the ITT, PP, and modPP populations. Box-Whisker plot showing median, first (Q1) and third (Q3) quartile, and mean value (X). Whiskers are 1.5 times inter quartile range (IQR = Q3–Q1)) or maximum/minimum values when these are within the 1.5 IQR range. Outliers (beyond the 1.5 times IQR) are presented as single dots. *p*-values are based on independent-samples t-test. DP: Dairy-based test product. Placebo: skimmed milk. PSQI: Pittsburgh Sleep Quality Index.

**Figure 3 nutrients-13-02204-f003:**
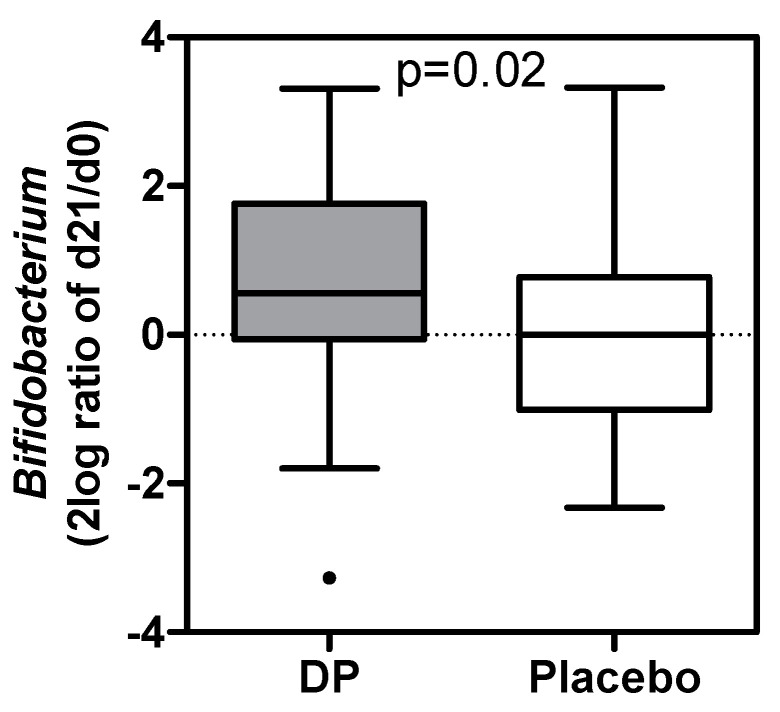
Bivariate analyses show an increase in *Bifidobacterium* over time (2log ratio of relative abundance at day 21 and day 0) for the DP and no change for placebo group (ITT) (*p* = 0.02). Boxplots are displayed as Tukey whiskers. DP: Dairy-based test product. Placebo: skimmed milk.

**Table 1 nutrients-13-02204-t001:** Composition of study products.

Nutrient	Unit	DP ^4^	Placebo
Amount per sachet ^1^	g	19	12
Protein	g	10	3.62
whey protein	g	9.5	0.72
casein	g		2.90
tryptophan	mg	500	75
other LNAA ^1^	mg	2930	573
tryptic casein hydrolysate ^2^	mg	200	
Lactose	g	1.9	5.4
GOS ^3^	g	5.2	
Magnesium	mg	200	13.1
Zinc	mg	5	0.12
Vitamin B_6_	mg	1	0.03
Niacin	mg	10	0.10
Vitamin D_3_	µ	10	<0.15

^1^ Amounts per sachet are different between DP (dairy-based test product) and Placebo to realize a comparable visual filling of the sachet. The skimmed milk powder (placebo) is less dense than the test product DP. The content of a sachet was dissolved daily in 150 mL lukewarm water and consumed about 1 h before going to bed. ^2^ Other large neutral amino acids: tyrosine, phenylalanine, leucine, iso-leucine, valine (Markus CR, Firk C, Gerhardt C, Kloek J, Smolders GF. Effect of different tryptophan sources on amino acids availability to the brain and mood in healthy volunteers. Psychopharmacology 2008;201:107-114). ^3^ Lactium^®^, 73% protein. ^4^ Galacto-oligosaccharides (Biotis^TM^ GOS), 70% pure GOS.

**Table 2 nutrients-13-02204-t002:** General characteristics of study participants at the start of the study for the ITT, PP, and modPP study groups.

	ITT	PP	modPP
**n**	69	64	47
**Gender: female/male (n) [ratio]**	54/15 [3.6]	51/13 [3.9]	37/10 [3.7]
**Age (y)**	39.5 ± 6.3	39.1 ± 6.2	38.5 ± 6.4
**Height (m)**	1.71 ± 0.10	1.71 ± 0.09	1.71 ± 0.08
**Weight (kg)**	68.2 ± 11.2	67.3 ±11.0	68.2 ± 10.5
**BMI (kg/m^2^)**	23.1 ± 2.0	23.1 ± 2.0	23.3 ± 1.9
**PSQI (at baseline)**	10.3 ±2.8	10.2 ±2.7	11.5 ± 2.0 ^a^
**Normal time to bed (h:min)**	23:37 ± 01:11	23:39 ± 01:13	23:40 ± 01:14
**Normal time to rise (h:min)**	07:38 ± 00:55	07:39 ± 00:56	07:42 ± 00:59
**Daily caffeine containing coffee (cups, (min-max))**	1.7 ± 1.5 (0-6.3)	1.7 ± 1.5 (0-6.3)	1.8 ± 1.5 (0-6.3)
**Daily alcohol containing drink (servings (min-max))**	0.7 ± 0.7 (0-3.1)	0.6 ± 0.7 (0-3.0)	0.6 ± 0.6 (0-7)
**Watching TV 30 min before bedtime (n,%)**	25 (36.2)	24 (37.5)	17 (36.2)
**Watching computer 30 min before bedtime (n,%)**	4 (5.7)	4 (6.3)	3 (6.4)
**Using the telephone 30 min before bedtime (n,%)**	37 (53.6)	36 (56.3)	26 (55.3)
**Reading a book 30 min before bedtime (n,%)**	5 (7.2)	5 (7.8)	2 (4.3)
**Self-reported mood quality at rise (5-1: good-bad)**	2.7 ± 1.0	2.7 ± 1.0	3.0 ± 1.0
**Self-reported fitness quality at rise (5-1: good-bad)**	2.9 ± 1.0	2.9 ± 1.0	3.1 ± 1.0

ITT: intention to treat; PP: per protocol; modPP: modified per protocol; BMI: body mass index; PSQI: Pittsburgh Sleep Quality Index. ^a^ Significantly (*p* = 0.042) different from ITT and PP following ANOVA with Bonferroni post-hoc analysis.

**Table 3 nutrients-13-02204-t003:** Changes in delta-PSQI and absolute PSQI between and within product groups, taking intervention periods 1 and 2 together.

	Products		*p*-Value ^1^
	DP	Placebo	
**ITT**			
Change d0–d7	−0.72 ± 3.25	−0.62 ± 2.64	0.841
Change d0–d14	−0.93 ± 2.84	−0.41 ± 3.35	0.325
Change d0–d21	−0.46 ± 3.05	−0.36 ± 3.14	0.848
Time effect ^2,^ *p*-value	0.379	0.723	
Post-hoc analysis	>0.370	1.0	
PSQI d0	9.83 ± 2.90	9.91 ± 3.04	0.817
PSQI d7	9.24 ± 2.90	9.29 ± 3.43	0.863
PSQI d14	8.90 ± 2.61	9.51 ± 3.11	0.216
PSQI d21	9.36 ± 3.08	9.55 ± 3.02	0.717
Time effect ^2,^ *p*-value	0.072	0.389	
Post-hoc analysis	>0.068	>0.321	
**PP**			
Change d0–d7	−0.63 ± 3.32	−0.55 ± 2.49	0.841
Change d0–d14	−0.92 ± 2.93	−0.22 ± 3.24	0.325
Change d0–d21	−0.50 ± 3.01	−020 ± 2.99	0.848
Time effect ^2,^ *p*-value	0.429	0.574	
Post-hoc analysis	>0.487	>0.992	
PSQI d0	9.84 ± 3.00	9.89 ± 2.92	0.864
PSQI d7	9.37 ± 2.89	9.34 ± 3.38	0.920
PSQI d14	8.92 ± 2.67	9.67 ± 3.13	0.216
PSQI d21	9.34 ± 3.01	9.69 ± 3.06	0.717
Time effect ^2,^ *p*-value	0.101	0.511	
Post-hoc analysis	>0.117	>0.501	
**ModPP**			
Change d0–d7	−0.91 ± 3.10	−066 ± 2.60	0.673
Change d0–d14	−1.60 ± 2.53	−0.30 ±3.28	0.034
Change d0–d21	−1.17 ± 2.76	0.43 ± 3.17	0.227
Time effect ^2,^ *p*-value	0.228	0.682	
Post-hoc analysis	>0.448	1.0	
PSQI d0	10.85 ± 2.55	10.57 ± 2.68	0.609
PSQI d7	9.87 ± 2.97	9.91 ± 3.28	0.944
PSQI d14	9.26 ± 2.83	10.28 ± 2.85	0.085
PSQI d21	9.68 ± 3.07	10.15 ± 3.25	0.474
Time effect ^2,^ *p*-value	0.001	0.487	
Post-hoc analysis	0–14 d: 0.001 0–21 d: 0.055	>0.531	

Data are presented as mean ± SD. DP: dairy-based test product. Placebo: skimmed milk. ModPP: modified PP population. ^1^ Independent Samples Test. ^2^ One-way repeated measures ANOVA with Bonferroni post-hoc analysis.

**Table 4 nutrients-13-02204-t004:** DASS 42 outcomes (total and sub-scores) per treatment and taking intervention periods 1 and 2 together.

	Products		*p*-Value ^1^
	DP	Placebo	
**ITT**			
DASS total score d0	16.5 ± 15.4; 13.0 (13.5)	14.2 ± 10.7; 12.0 (13.5)	0.510
DASS total score d21	14.9 ± 13.1; 11.0 (16.5)	13.0 ± 11.5; 11.0 (14.0)	0.427
*Within group* ^2^ *; p*	0.150	0.038	
DASS stress d0	7.4 ± 5.9; 6.0 (8.0)	6.7 ± 4.5; 6.0 (7.0)	0.611
DASS stress d21	6.7 ± 4.8; 6.0 (7.0)	5.9 ± 4.8; 4.0 (7.0)	0.255
*Within group; p*	0.318	0.041	
DASS anxiety d0	3.1 ± 4.2; 2.0 (2.5)	2.4 ± 2.9; 1.0 (2.0)	0.291
DASS anxiety d21	2.7 ± 3.4; 2.0 (3.0)	2.1 ± 2.9; 1.0 (2.0)	0.178
*Within group; p*	0.374	0.116	
DASS depression d0	6.0 ± 7.3; 3.0 (6.0)	5.1 ± 5.1; 3.0 (6.0)	0.679
DASS depression d21	5.5 ± 6.7; 3.0 (6.5)	5.0 ± 5.3; 3.0 (7.0)	0.918
*Within group; p*	0.260	0.649	
**PP**			
DASS total score d0	17.0 ± 15.9; 14.0 (14.8)	14.3 ± 10.8; 12.0 (12.5)	0.426
DASS total score d21	14.7 ± 13.0; 11.0 (15.0)	13.1 ± 11.8; 10.5 (13.5)	0.492
*Within group; p*	0.055	0.055	
DASS stress d0	7.7 ± 6.0; 6.0 (6.0)	6.8 ± 4.5; 6.0 (6.8)	0.441
DASS stress d21	6.7 ± 4.7; 6.0 (6.8)	5.9 ± 4.8; 4.5 (7.0)	0.316
*Within group; p*	0.125	0.044	
DASS anxiety d0	3.2 ± 4.3; 2.0 (3.0)	2.5 ± 2.9; 1.0 (2.0)	0.304
DASS anxiety d21	2.7 ± 3.3; 1.5(2.8)	2.2 ± 3.0; 1.0 (2.0)	0.224
*Within group; p*	0.325	0.187	
DASS depression d0	6.1 ± 7.5; 3.0 (7.5)	5.0 ± 5.2; 3.0 (6.0)	0.694
DASS depression d21	5.3 ± 6.7; 3.0 (5.0)	5.0 ± 5.4; 2.5 (6.8)	0.927
*Within group; p*	0.102	0.810	
**modPP**			
DASS total score d0	19.1 ± 17.2; 16.0 (16.0)	15.8 ± 11.3; 13.0 (13.0)	0.382
DASS total score d21	16.4 ± 13.9; 11.0 (19.0)	14.4 ± 12.1; 12.0 (17.0)	0.633
*Within group; p*	0.065	0.098	
DASS stress d0	8.5 ± 6.3; 8.0 (6.0)	7.5 ± 4.2; 7.0 (6.0)	0.521
DASS stress d21	7.3 ± 4.5; 7.0 (6.0)	6.4 ± 5.0; 5.0 (6.0)	0.275
*Within group; p*	0.131	0.048	
DASS anxiety d0	3.5 ± 4.7; 2.0 (3.0)	2.6 ± 3.2; 2.0 (2.0)	0.319
DASS anxiety d21	3.0 ± 3.7; 2.0 (5.0)	2.2 ± 3.0; 1.0 (2.0)	0.206
*Within group; p*	0.522	0.094	
DASS depression d0	7.1 ± 8.2; 4.0 (8.0)	5.6 ± 5.5; 3.0 (7.0)	0.487
DASS depression d21	6.1 ± 7.4; 4.0 (10.0)	5.8 ± 5.6; 4.0 (9.0)	0.706
*Within group; p*	0.067	0.674	

Data are presented as mean ± SD; median (IQR). ^1^ Mann–Whitney U test. ^2^ Related Samples Wilcoxon Signed Rank test. DP: dairy-based test product; Placebo: skimmed milk powder; d0: baseline; d21: 21 days of intervention.

**Table 5 nutrients-13-02204-t005:** Early morning saliva cortisol levels at baseline (day 0) and after 21 days of intervention (day 21) per treatment and in ITT, PP and modPP populations.

	Period 1 + 2		*p*-Value ^1^
	DP	Placebo	
**ITT**			
Cortisol d0, 0 min	3.36 ± 1.54; 3.20 (2.20)	3.38 ± 1.44; 3.05 (2.18)	0.973
Cortisol d21, 0 min	3.19 ± 1.45; 3.00 (2.20)	3.78 ± 1.78; 3.50 (1.85)	0.045
*Within group; p*	0.316	0.161	
**PP**			
Cortisol d0, 0 min	3.38 ± 1.56; 3.20 (2.10)	3.34 ± 1.44; 3.00 (2.05)	0.784
Cortisol d21, 0 min	3.21 ± 1.45; 3.20 (2.20)	3.83 ± 1.82; 3.50 (1.95)	0.059
*Within group; p*	0.414	0.098	
**modPP**			
Cortisol d0, 0 min	3.38 ± 1.64; 3.10 (1.90)	3.38 ± 1.46; 3.10 (2.15)	0.788
Cortisol d21, 0 min	3.03 ± 1.36; 2.90 (2.10)	3.90 ± 1.91; 3.60 (2.05)	0.033
*Within group; p*	0.248	0.170	

Data are presented as mean ± SD; median (IQR). ^1^ Mann-Whitney U-test. DP: dairy-based test product; Placebo: skimmed milk powder; d0: baseline; d21: 21 days of intervention.

## Data Availability

The raw 16S rRNA gene sequencing data are available from the European Nucleotide Archive under accession number PRJEB45898. . Other clinical data presented in this study are available upon substantiated request from the corresponding author. The data are not publicly available due to not having that service in place (yet).

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
