# Peer review of "The Effect of A Whey-Protein and Galacto-Oligosaccharides Based Product on Parameters of Sleep Quality, Stress, and Gut Microbiota in Apparently Healthy Adults with Moderate Sleep Disturbances: A Randomized Controlled Cross-Over Study"

_nutrients, 2021, doi:10.3390/nu13072204_

Round 1

Reviewer 1 Report

Dear Authors,

This is an interesting manuscript that is investigating one of the important public problems, it seems that the number of people affected by sleep disturbances is constantly increasing with one of the possible causes being our 'modern live. Please find several comments below. 

Methods

Line 105 - please make it clear how the slice samples were collected and stored; was it a self-collection and stored by participants? 

Line 109 - the same as above

Line 114 - how did you ensure that the samples of saliva and faecal matter were collected according to the protocol and that they were stored correctly? 

Line 139 - how was compliance monitored? just empty sachets? 

Reviewer 2 Report

This paper deals with the potential effect of a dairy-based product containing tryptophan, galacto-oligosaccharides, and a casein hydrolysate as bioactive components or ingredients, on sleep quality. The results show that there is no effect on sleep quality and the authors found an effect on the microbiota and on next morning salivary cortisol that is assumed to have a relation with sleep quality. This association is not proved and thus the conclusion should not be taken for granted.

We would state from the title that this is the case and that the dairy-based product had no effect on sleep quality.

In the abstract the use of abbreviations like ITT, PP and mod-PP for the groups studied in the statistical analysis should be avoided and a clear statement should be used instead. The abstract should be clear and explain the results so the reader can understand from the abstract what are the real results from this study. 

Reading the paragraph starting with "Stimulation of orexin..." and the following one it is not clear what is the relation between oexin levels, serotonin levels and sleep quality. Could you please be more clear or if it is a controversial subject and it is not clear what is the association between sleep quality and levels of blood glucose, insulin, serotonin and gamma-aminobutyric acid (GABA), then say so.

Is this true? "tryptophan (Trp)... is the precursor of serotonin (relaxing neurotransmitter) which is used for melatonin (sleep supporting hormone) production. Please give a reference for this

The fact the control in the diet is skimmed milk is very important and should be included from the abstract. 

the link https://www.telepsy.nl/dass does not exist anymore or does not open. Could you please, if you think it is necessary to include it, include in the supplementary file the actual questionnaire that was used? or avoid the link

It is not clear from the abstract but in the methodology we understand that microbiota was only studied for the first arm in the intervention, and not sure for the other parameters cannot be included as a result from the full study, maybe indicate the n between brackets in the abstract when talking about the microbiota effects.

It is difficult to get to conclusions when losing almost 25% of the group because of the differences in PSQI values differed so strongly from inclusion day to baseline day. We understand that the different data sets were done in order to find a statistical significant difference between treatments but we are not sure this is realistic.

Why the within group ps are not included in table 5? could be included

The effect of skimmed mild on cortisol levels need to be commented and an explanation for it should be given. It is also very interesting the fact that cortisol levels are so similar between groups ate baseline (almost the same exact value) and so different after treatment. How do you explain this? Can there be an effect of the SmartSleep devise itself on the quality of sleep? Please comment on this.

The conclusions, title and abstract need to be changed to make clear that in fact there was not effect at all of DP and if any there is an increase of first morning salivary cortisol levels for the skimmed milk intervention (control or placebo). Saying that DP decrease the value in competition with the placebo is not totally true looking at the results.
